# Gradient Microstructure Induced by Surface Mechanical Attrition Treatment in Grade 2 Titanium Studied Using Positron Annihilation Spectroscopy and Complementary Methods

**DOI:** 10.3390/ma14216347

**Published:** 2021-10-23

**Authors:** Konrad Skowron, Mirosław Wróbel, Michał Mosiałek, Léa Le Joncour, Ewa Dryzek

**Affiliations:** 1Institute of Nuclear Physics Polish Academy of Sciences, PL-31342 Kraków, Poland; ewa.dryzek@ifj.edu.pl; 2Faculty of Metals Engineering and Industrial Computer Science, AGH University of Science and Technology, al. A. Mickiewicza 30, PL-30059 Kraków, Poland; mwrobel@agh.edu.pl; 3Jerzy Haber Institute of Catalysis and Surface Chemistry, Polish Academy of Sciences, Niezapominajek 8, PL-30239 Kraków, Poland; nbmosial@cyfronet.pl; 4Université de Technologie de Troyes (UTT), LASMIS, 12 rue Marie Curie, 10010 Troyes, France; lea.le_joncour@utt.fr

**Keywords:** titanium, surface mechanical attrition treatment, SMAT, gradient structured materials, positron annihilation spectroscopy, corrosion

## Abstract

Microstructural changes in grade 2 titanium generated by surface mechanical attrition treatment (SMAT) were studied using positron annihilation lifetime spectroscopy and complementary methods. A significant increase in the mean positron lifetime indicated many lattice defects introduced by SMAT. Two positron lifetime components were resolved in the positron lifetime spectra measured. The longer lifetime revealed the presence of vacancy clusters containing about 3 or 4 vacancies, while the shorter one was attributed to the annihilation of positrons trapped at dislocations. The changes of the positron lifetime indicated a decreasing dislocation density and the presence of a deeper layer with a higher concentration of vacancy clusters at the distance from the treated surface for which the microhardness approached the value for the strain-free matrix. Electrochemical impedance spectroscopy showed the positive effect of SMAT on the corrosion resistance of the titanium studied in a saline environment also after removal of the original oxide layer that was formed during the SMAT.

## 1. Introduction

There is a growing interest in nanocrystalline (NC) materials [1], or gradient materials with greatly refined structures of a subsurface layer. These materials, as well as methods of their production attract the attention of researchers due to their existing and potential applications in almost every technological area. In particular, methods based on severe plastic deformation (SPD) have become popular presently due to their simplicity and ease of applicability for plentiful classes of materials [2]. The application of these methods results in the creation of ultrafine-grained structure, as well as a great number of crystal lattice defects which introduction greatly changes the properties of the material. The ultrafine-grained materials can be produced by most SPD techniques. However, NC materials can be obtained only by methods that are based on non-homogeneous deformation with large strain gradients [3].

Various techniques based on SPD that modify the entire volume of the processed material, i.e., bulk nanocrystallization methods, e.g., equal channel angular pressing (ECAP) or hydrostatic extrusion (HE), as well as, those based on the surface self nanocrystallization (SSN) have been developed and their popularity continues to grow [4]. They are performed without changing the chemical or phase composition of the substrate. SSN is much cheaper and easier to carry out in comparison to bulk nanocrystallization. During the treatment, only the top layer of the base material experiences microstructural changes. The nanostructured surface usually enhances physical properties such as hardness, fatigue strength, tribological properties and can promote the formation of a passivation layer that is more corrosive resistant in comparison to conventional coarse-grained metals [5,6]. The methods of SSN include, e.g., high-speed rotation wire-brushing (HRWB) or shot peening (SP), technologies such as air blast (ABSP), ultrasonic shot peening (USP), or surface mechanical attrition treatment (SMAT) [7,8]. It should be noted that according to Grosdidier and Novelli, the latter two methods share the same concept especially in the case of ultrasonic SMAT [9].

SMAT is the SPD method that can produce a hard NC layer and gradient microstructure in the treated surface [5]. The method shares the same features with conventional ABSP, i.e., the surface microstructure modification is induced by repeated impacts of high-velocity spherical shots with a few millimeters diameter on the treated surface. In both methods, the shots randomly hit the sample surface causing plastic deformation of the material. However, in SMAT, the shots and the processed material are placed in a closed chamber and are vibrated with a very high frequency using a vibration generator. The vibration frequency of the chamber is in the range of 50 Hz to 20 kHz [10]. Multidirectional impacts induce grain refinement to the nanometer scale near the target surface. SMAT generates many defects such as dislocation and deformation twins and consequently refines the topmost layer of the sample to the nanometer scale. SMAT introduces local SPD to a target surface without changing the properties of the whole volume of the workpiece hence it requires less energy in comparison to the methods of bulk nanocrystallization. For example, ECAP requires the use of large loads and specially designed dies [11]. Due to this fact, SMAT can be considered environmentally friendly [12]. The formation of a nanostructured layer as a result of SMAT has been reported for many metals and alloys such as, e.g., stainless steel, aluminum, copper, titanium [13,14,15,16,17,18,19,20]. Due to its simplicity and flexibility, SMAT can be potentially used for commercial purposes. For example, it was found that SMAT greatly reduces the nitriding temperature of iron [21]. It can be used as a pretreatment for anodizing of Ti [22], or to assist the formation of nanoporous titania by chemical oxidation and calcination [23]. According to published papers on SMAT applications in biomedicine, there are reports that the method enhances, e.g., cell attachment, differentiation, and osseointegration of titanium-made implants [24,25]. However, further studies are needed before SMAT may be clinically used.

Properties such as high strength, low density, and excellent corrosion resistance make titanium useful in many potential applications. Ti-based materials have been used in such areas as aerospace and automotive, power generation (turbine engines), chemical processing, and biomedical devices [26,27,28,29]. Titanium and its alloys are considered among the best materials for hard-tissue implants due to the combination of their mechanical properties, good biocompatibility, and osseointegration [30,31]. The most used titanium-made materials are commercial purity (cp) Ti and Ti-6Al-4V alloy. Nonetheless, the wear resistance of titanium is relatively low which limits the potential area of its use in load-bearing applications, e.g., in articulating components of total joint replacements. Mechanical and chemical properties of titanium-based materials can be improved with appropriate mechanical treatments, e.g., SMAT [32].

The microstructure created by the application of SMAT is usually investigated using standard experimental techniques, e.g., optical, scanning, and transmission electron microscopy, X-ray, and electron diffraction [20,33]. In the present study positron annihilation spectroscopy (PAS) is applied for this purpose. PAS is a suitable tool for qualitative characterization of open volume defects which are created during plastic deformation induced by various mechanical treatments [34,35]. In general, this technique is non-destructive and extremely sensitive to the presence of crystal lattice defects in the material studied. Thus, it allows tracing microstructural changes in a material after plastic deformation.

This paper aims to characterize the microstructure of grade 2 titanium subjected to SMAT and to relate the obtained results to corrosion properties, determined by electrochemical tests. Positron annihilation lifetime spectroscopy (PALS), X-ray diffraction (XRD), electron backscatter diffraction (EBSD), microhardness tests, and the surface profile characterization were applied to characterize the influence of SMAT on the microstructure of the Ti surface.

Noteworthy is the fact that PAS can shed light on the complicated mechanism of deformation induced by SMAT in the surface layer of titanium by providing information on crystal lattice defects created in this layer.

## 2. Materials and Methods

The specimen 100 mm × 100 mm × 8.3 mm in dimensions was cut from a commercial sheet of titanium grade 2 (ADMETAL, Chwaszczyno, Poland). Prior to SMAT, the specimen was annealed at 400 °C in a vacuum (~10^−3^ Pa) for 0.5 h, and then cooled to room temperature within the furnace. The purpose of the annealing was to obtain the material with only residual crystal lattice defects. SMAT was performed in the air at room temperature using the set-up with the stainless-steel chamber and 2 mm stainless-steel balls. The vibration frequency was 20 kHz. Each of the two sides of the titanium plate was treated separately for 60 s with the vibration amplitude of 13 ± 2 µm and 120 s with the vibration amplitude of 27 ± 2 µm, respectively. They are referred to as SMAT-1 and SMAT-2. An increase in both the SMAT duration and the vibration amplitude significantly increased the energy imparted into the treated surface. The vibration amplitude influences the normal speed of the impacts. In this case, doubling of the vibration amplitude doubles the velocity of the steel balls which leads to their approximately four-time higher kinetic energy [36]. After SMAT, the plate was cut into smaller samples 10 mm × 10 mm in size. The untreated rest of the plate was kept as a reference.

For XRD, a PANalytical Empyrean diffractometer (Malvern Panalytical Co., Almelo, The Netherlands and Malvern, UK) with Cu Kα radiation was applied. The measurements were carried out using a parallel beam geometry (Göbel mirror in the incident beam optics and parallel plate collimator in the diffracted beam optics) over the 2θ range of 20–120°, step 0.02°, at room temperature.

A FEI Nova NanoSEM 450 scanning electron microscope (SEM) (FEI Co, Hillsboro, OR, USA) was used for the microstructure observations on the cross-section of the sample SMAT-2. The sample was mounted in cold mounting resin. The further sample preparation consisted of grounding with a series of SiC papers down to grid 2400, and polishing using aluminum oxides with grain size down to 0.05 µm and 0.04 µm silica oxide. Finally, it was cleaned with an argon ion beam using an IM4000Plus instrument (Hitachi High-Tech Corporation, Tokyo, Japan).

The microhardness measurements were performed using a 2500 Instron ITW tester (Instron, Norwood, MA, USA) with a Knoop indenter with a maximum load of 0.01 kg (HK0.01). During measurements, the intender longer diagonal was parallel to the trace of the treated surface. All recommendations of the ISO 4545 standard were respected. The surface roughness was measured using an optical profiler WYKO NT9300 (Veeco, New York, NY, USA).

PALS measurements were carried out using the digital spectrometer manufactured by TechnoAP (TechnoAP Co., Ltd. Mawatari, Hitachinaka-shi, Ibaraki, Japan) with BaF_2_ scintillators coupled to photomultipliers H3378-50 (Hamamatsu, Shimokanzo, Iwata City, Japan). The time resolution of the spectrometer was about 200 ps (full width at half maximum). During the measurement, the source of the positrons containing ^22^Na isotope enveloped into a 7 µm thick Kapton foil was used. The source sandwiched between two SMATed titanium samples was placed in the spectrometer. All the measured spectra with more than 10^6^ counts were analyzed using the LT program including the source contribution and background subtraction [37].

Positrons emitted from the radioactive ^22^Na isotope probe a certain depth inside the sample they are implanted to. The linear absorption coefficient *µ* for positrons from the ^22^Na source implanted in titanium is equal to ca. 222.4 µm^−1^ [38]. Accordingly, the contribution to the measured positron lifetime value derives from a layer of a thickness corresponding to 1/µ. Thus, the mean implantation depth of positrons is c.a. 45 µm, while the SMAT affected layer is much deeper. Hence, it is possible to obtain depth profiles of positron annihilation characteristics, e.g., the mean positron lifetime. The depth profiles of the positron lifetime were obtained by sequential removal of subsequent layers from the sample surface by etching in an acidic solution of 85%H_2_O + 10%HNO_3_ + 5%HF. After each etching, the thickness of the removed layer was measured using a digital micrometer caliper with ±1 µm accuracy and PALS measurements were carried out. It is assumed that the chemical removal of the sample layers does not affect the measured positron lifetime values [39,40].

Two series of measurements were performed during corrosion resistance tests. In the first one, the surfaces of the SMATed and the reference samples were only cleansed with acetone, and then electrochemical impedance spectroscopy (EIS) measurements were performed. To facilitate the description of the results of corrosion resistance measurements, these specimens are referred to as as-received ones. In the second series, Ti pieces after SMAT and the reference ones were grounded and polished to remove the presumably thick oxide layer that had grown on their surfaces. SiC papers of a grid 1000, 2000, 2400 were used. Then, the samples were further polished using paper of a grid 4000 and then cleansed with acetone. The thicknesses of abraded layers were determined by measuring the dimensions of the samples and their weight loss after the polishing process. Approximately 2 µm thick layers were abraded in each polished sample. The specimens from the second series are referred to as polished ones. Samples SMAT-1 and -2, as well as the reference ones, were measured in each of the series. At least three measurements were made for each sample.

EIS was carried out in an all glass-and-Polytetrafluoroethylene (PTFE) cell in a 0.15 mol dm^−3^ NaCl solution. The composition of the solution used is close to the solutions in which Ti objects corrode in practical applications, e.g., implants in the human body and land transportation, e.g., sodium chloride is spilled on the road surfaces during winter.

A three-electrode corrosion testing set-up consisting of a saturated calomel electrode as the reference electrode (RE), platinum foil as the counter electrode, and the sample as the working electrode (WE), was used. All potentials are reported versus RE. The WE with an exposed area of 0.25 cm^2^ was placed at the bottom of the cell, in a horizontal position, with the active surface up to facilitate the eventual liberation of hydrogen formed in the titanium passivation process. Before the measurements, the solution was bubbled with argon to remove dissolved oxygen. The measurements were carried out at room temperature 21 ± 1 °C. All titanium EIS measurements were preceded by a 24-h immersion in the solution accompanied by simultaneous measurement of the open circuit potential (OCP). EIS spectra in the frequency range of 225 µHz to 300 kHz and the amplitude of the sinusoidal voltage signal of 10 mV were registered at OCP with the density of 8 points per decade using Gamry G300 Potentiostat/Galvanostat/ZRA (by Gamry Instruments, Warminster, PN, USA).

## 3. Results and Discussion

### 3.1. Surface Characterization

It is known that the surface roughness of titanium implants is an important factor influencing their rate of osseointegration and stability [41,42]. Figure 1 presents the surface morphology of titanium samples SMAT-1 and 2. The surface of the sample SMAT-1 appears more homogenous. On the surface of the sample SMAT-2, there are visible deeper craters formed due to the significantly higher impact energy of steel shots. This difference is reflected in the values of the roughness parameter *R*_a_. It is equal to 1.65 µm and 3.07 µm for SMAT-1 and -2 specimens, respectively, while the roughness of the reference sample without treatment was close to 1 µm. The increase of the surface roughness is in agreement with the results reported by Zhu et al. who studied the influence of process parameters of ultrasonic shot peening on surface roughness and hardness of cp titanium [43,44]. Those authors found that the surface roughness of cp titanium increases with the peening duration at the beginning of the process and then saturates. However, the influence of shots kinetic energy due to an increase in the sonotrode amplitude is more pronounced than the influence of the treatment duration. The increase in the sonotrode amplitude leads to larger and deeper pits which are visible for the SMAT-2 sample with twice the sonotrode amplitude. In the case of magnesium SMATed with the same parameters as in the present studies, the differences in the surface roughness and morphology between the samples such treated were smaller [45].

### 3.2. XRD Results

To determine the effect of SMAT on the structure of the topmost layer of the samples the broadening of the XRD peaks was measured in the symmetrical diffraction mode. The average crystallite size was calculated using the Williamson-Hall method [46]. All registered peaks from the XRD patterns were used in the analysis. The obtained values of the crystallite sizes and the lattice microstrain are gathered in Table 1. The difference in the crystallite size between the two samples studied is negligible if the uncertainties of the obtained values are taken into account. However, the samples differ in the microstrain values. The microstrain for the SMAT-2 sample is visibly higher than that for the SMAT-1 one.

Generally, due to the high strain and strain rate during SMAT, the grains near the treated surface have sizes well below 100 nm while other bulk severe plastic deformation techniques produce grains of typical size greater than 100 nm [47]. However, when comparing the obtained values of the crystallite size determined by X-ray diffraction with the grain sizes determined by transition electron microscopy (TEM), it should be remembered that the latter are usually equal to or greater than the former, as reported, e.g., for SMATed cp titanium by Bahl et al. [48]. The crystallite size in the present study is comparable to those reported in the literature obtained using XRD measurements even for different SMAT parameters [49,50]. They are larger than those obtained by Agrawal et al. for cp titanium and similar SMAT parameters including the short duration between 30 and 120 s, which were close to 20 nm with the microstrain close to 0.20% [51]. However, there is even a better concurrence between the present results and those reported in ref. [44] also obtained for similar SMAT parameters for cp titanium.

### 3.3. Microstructure Analysis

The microstructure of the SMATed titanium was observed using the EBSD technique. The image of the microstructure of the cross-sectioned titanium sample SMAT-2 is shown in Figure 2. The SMATed surface is located on the bottom edge of the picture. However, it should be noted that the edge of the specimen is not visible, due to its undesirable rounding during preparation made before measurement. The EBSD scan was recorded starting at approximately 5 µm below the SMATed surface. The area rich in refined grains is visible up to the depth of about 50 µm. The inverse pole figures show that close to the surface, no significant preferred orientations of the grains were developed. Deeper, small grains clusters were formed. These clusters are usually located along lines inclined concerning the treated surface and their grains have some preferred crystallographic orientation. A more resolving research technique (e.g., TEM) is required for the precise determination of the preferred matrix and grains orientation relationship. No twins were observed in small grains very probably because, in small grains, a twin can completely consume and reorient a parent grain similarly as observed for the alloy Ti-6Al-4V [52].

### 3.4. Microhardness Measurement

Figure 3 shows the variation of the microhardness with the distance from the treated surface. For the SMAT-1 sample, the increase in the microhardness at the surface is relatively small. The microhardness decreases and reaches the initial reference value at a distance of about 50 µm. The increase of the microhardness at the surface for the sample SMAT-2 is much higher. The microhardness decreases almost linearly to the distance of about 100 µm. This corresponds to the decreasing density of refined grains in Figure 2. The points closest to the surface measured at a distance of about 10 µm may indicate a presence of a harder top layer. Deeper than 100 µm, where the microhardness approaches the reference value, its decrease is slower, and the measurement points are more scattered. The microhardness measurements show that the whole layer affected by SMAT extends to the depth of about 200 µm. The measured dependencies confirm the results of the surface roughness measurements and microstrain assessment indicating that plastic deformation of the surface layer of the SMAT-2 sample is much higher than the SMAT-1 treated with the lower sonotrode amplitude. It agrees with the results reported by Zhu et al. [44]. According to their findings, the surface hardness of cp titanium increases with the increase of the sonotrode amplitude and treatment duration. In the latter case, after the initial increase, the hardness tends to be stable after a certain SMAT duration. Both the kinetic energy of shots and impact coverage determine the amount of energy imparted into the treated surface and hence plastic deformation in the surface layer.

The mechanism of nanocrystalization of the surface region and generation of the gradient microstructure in titanium has been studied earlier [53,54]. During grain refinement by SMAT, titanium is deformed by a combination of slip and mechanical twinning, and it was found that the prism slip and twinning are the main deformation modes in the first period of the surface treatment. As the strain level increases the basal slip systems can be activated. For titanium, the critical resolved shear stress for the prism slip systems are lower than for the basal ones [55]. Such deformation mode results in creating nanograins at the treated surface. Since the applied strain and strain rate gradually decrease with the increasing distance from the treated surface a transition layer containing deformed macrograins forms below the fine-grain layer. The twin density increases with the grain size of deformed material, in agreement with the literature [56] The transition layer is followed by the strain-free matrix. The decrease in the microhardness with the increasing distance from the treated surface reflects the gradient microstructure resulting from a gradual decrease in the applied strain and strain rate.

### 3.5. Positron Lifetime Measurements

Two components in the positron lifetime spectra, described by their lifetime values *τ_i_* and intensities *I*_i_ for i = 1, 2, were observed for both samples in the etching experiment. Figure 4 presents the dependencies of the lifetime values of these components and the intensity of the longer component *I*_2_ on the distance from the SMATed surface. Additionally, there is shown the dependence of the mean positron lifetime in Figure 5. The mean positron lifetime is a commonly accepted robust parameter that does not depend on the spectrum decomposition into lifetime components. It is defined as follows:(1)$$\bar{\tau }=I_{1}\tau_{2}+I_{2}\tau_{2}$$
where *I*_1_ *+ I*_2_ = 1.

At the surface, $\bar{\tau }$ takes similar values for both samples. For the SMAT-1 specimen, the decrease in the mean positron lifetime is initially slow to the distance of 75 µm, then it is steeper to 150 µm. The mean positron lifetime reaches the reference value for the annealed titanium at 250 µm. The reference value is the positron lifetime for well-annealed titanium grade 2 which in our case is equal to 144 ps. It can be compared with the literature values, e.g., 144.6 ps for titanium (99.7% purity) reported by F. Lukáč et al. [57]. The depth of 250 µm can be treated as the total range of defects induced by SMAT. The mean positron lifetime behavior for the SMAT-2 specimen is slightly different. After an initial decrease, which takes place to the depth of about 25 µm, the mean positron lifetime increases slightly reaching a maximum at 180 µm. Then it goes down to the reference value at 400 µm.

Analysis of the behavior of the positron lifetime components shown in Figure 4 indicates that for the SMAT-1 sample, the shorter lifetime, *τ*_1_, decreases rapidly starting from the value of about 170 ps at the surface to the values only slightly higher than the reference lifetime already at the depth less than 100 µm. The longer lifetime, *τ*_2_, takes the values close to 300 ps while its intensity, *I*_2_, increases starting from the surface reaching a maximum of 35% at the depth of 30 µm. Then it decreases and disappears deeper than 200 µm.

For the SMAT-2 specimen, the values of *τ*_1_ close to 170 ps or 180 ps are visible to the depth of about 150 µm. Then *τ*_1_ decreases steeply to the values close to the reference lifetime. The longer lifetime, *τ*_2_, takes the values close to 300 ps or slightly lower ones, after exhibiting some higher values in the vicinity of the surface where its intensity is low. The intensity, *I*_2_, starts to increase at the depth of 90 µm reaching a maximum of 35% at about 220 µm. After that, *I*_2_ decreases and disappears deeper than 350 µm. The shorter component with a lifetime between 170 ps and 180 ps can be ascribed to the annihilation of positrons trapped at dislocations [57,58]. The second component with lifetime τ_2_ close to 300 ps is higher than the positron lifetime in vacancies, i.e., 222 ps reported by Kaupilla et al. [59]. It originates from the annihilation of positrons trapped at vacancy clusters containing about 3 or 4 vacancies according to the theoretical calculations performed by Čížek et al. [60].

Similar values of the positron lifetime components were obtained by Lukáč et al. for pure titanium prepared by high-pressure torsion (HPT). In that case, the long lifetime, of 280 ps was slightly lower in comparison to the present results and its intensity was about 10% [57]. For the Ti-6Al-7Nb alloy also processed by HPT, Janeček et al. obtained the long lifetime coming from positron annihilation in vacancy clusters close to 300 ps with an intensity also about 10% [58].

Taking all this into account, it can be stated that SMAT introduces a gradient structure of defects in the affected layer. For the SMAT-2 sample, saturated trapping of positrons in dislocations and vacancy clusters takes place to the depth at which *τ*_1_ starts to decrease taking values below about 170 ps. However, the contribution of these two positron traps may change with the depth. The higher values of the longer positron lifetime close to the surface indicate that the vacancy clusters there are larger than in deeper layers. It should be noted that the microhardness decreases with the increasing depth for this region which indicates that the dislocation density also decreases. Nonetheless, the density of dislocations and vacancy cluster concentration is still high enough to cause trapping of all positrons. For the SMAT-2 sample, this layer is about 80 µm thick. For the SMAT-1 sample, the layer with saturated trapping of positrons in dislocations and vacancy clusters, if present, is only a few micrometers thick. For both samples, there is visible another layer with a vacancy cluster concentration higher than in the adjacent areas. It is placed deeper inside the material and extends to the end of the whole range of the observed changes. Moreover, the maximum of the vacancy cluster concentration indicated by the maximum of *I*_2_ occurs at the depth for which *τ*_1_ decreases to the reference lifetime indicating a significantly reduced dislocation density. The increase of the SMAT time and the sonotrode amplitude shifts this vacancy cluster layer deeper into the material which is connected to the increasing total depth of the whole affected layer.

During SMAT, the gradient microstructure is formed due to a gradient variation of the strain and strain rate from the treated top surface, where both are large, to the matrix material, where they are essentially zero. Strain rates as high as 10^4^ are developed during SMAT [61,62]. In comparison, the strain rate of other SPD methods such as HPT and ECAP is small or medium despite large strain. Even though the high strain rate deformation during SMAT is not considered to be shock deformation process it seems interesting to refer to the results of the PALS measurements of shock-loaded titanium for which the strain rate is extremely large. The similarities in the mode of deformation of SMAT and oblique impact/shock-wave loading were noticed by Bahl et al. in the case of microband formation in 316L stainless steel during SMAT [63]. For shock-loaded titanium, two components in the PALS spectra were revealed as well [64]. The intensity of the component originating from annihilation in vacancy clusters is comparable with the maximum intensity obtained in the present studies, which is higher than those reported for titanium deformed by HPT [57]. However, for the SMATed samples, the maximum intensity of the longer lifetime component occurs at some distance from the SMATed surface for which much lower strain and strain rate than that at the surface are expected. At the distance from the treated surface at which this maximum occurs, the microhardness already approaches the reference value.

The mean positron lifetime values similar to those for the SMATed titanium were obtained for pure titanium subjected to dry sliding against stainless steel [65]. It is known that sliding contact is accompanied by severe plastic deformation localized within a small volume of material adjacent to contact surfaces. The distribution of local strain and strain gradient in the deformed subsurface zone are connected to the distribution of crystal lattice defects beneath the worn surface [65]. In general, for both processes: SMAT and sliding friction, a gradient of the defect concentration occurs in the subsurface region. The mean positron lifetime profiles for the SMATed samples exhibit some similarities to the profiles in pure titanium after dry sliding, i.e., the layer with a high value of the mean positron lifetime above 190 ps extending to some depth beneath the surface and a second layer where the mean positron lifetime decreases to the reference value for the strain-free matrix. Similar to the SMATed sample, the longer lifetime component exhibits a high intensity of about 40% at the depth at which the shorter lifetime approaches or is lower than the reference value.

Basing on these similarities, the dependences of  shown in Figure 5 were described using a simple sigmoidal formula as proposed in ref. [66]:(2)$$\bar{\tau }=\tau_{0}+\frac{a}{2}\left[1+erf\left(-\frac{z-z_{0}}{\sqrt{2}b}\right)\right]$$
where *z* is the depth from the surface, and *τ*_0_, *z*_0_, *a*, and *b* are the adjustable parameters which values can be obtained in the fitting procedure. The *z*_0_ parameter is the depth at which the sigmoidal curve reaches the transition center. The dashed lines in Figure 5 present the results of the best fits of Equation (2) to the experimental points. It can be seen that the fitted dependencies describe the experimental data reasonably well, particularly for the SMAT-1 sample. For the sample SMAT-2, there is a discrepancy in the region of a small maximum for the depth of 180 µm. Table 2 contains the values of the fitted parameters together with the values of the total depth of the SMAT affected region, i.e., the depth where the reference value of the mean positron lifetime is reached (Figure 5). The total depth and the *z*_0_ parameter values increase with the SMAT time and the sonotrode amplitude. There is also a difference in the values of the *b* parameter. The comparison with the samples subjected to sliding indicates that the shape of the mean positron lifetime dependency is similar, but the defects are induced by SMAT at a greater distance from the surface even for as short treatment as 60 s.

### 3.6. Corrosion Tests

The Equivalent Electrical Circuit (EEC) commonly used for fitting the impedance data for Ti in saline solutions is shown in Figure 6 [67]. The model used assumes that the passive film on the titanium surface composes of two layers: an outer porous layer and an inner barrier layer. In the present model *R*_s_, *R*_p_, and *R*_b_ represent solution resistance, the porous layer resistance, and the inner barrier layer resistance, respectively. The barrier layer and the double layer of the oxide/solution interface capacitances are denoted as *Q_b_* and *Q_p_*, respectively [68]. Constant phase elements (CPEs) were used instead of pure capacitors in the fitting routine. The impedance *Z*_CPE_ of CPE can be expressed by [68]:(3)$$Z_{\mathrm{CPE}}=\frac{1}{2\pi f_{0}C_{0}}{\left(\frac{f_{0}}{if}\right)}^{\alpha }$$
where *i* is the imaginary unit, *f*_0_ is the frequency of reference (assumed 1000 Hz), *f* is the frequency, *C*_0_ is the capacitance at the frequency of reference, and *α* is a coefficient which is close to 1 for an ideal capacitor and it usually ranges between 0.7 and 1.0 in the case of a non-ideal capacitance [68]. For the EEC showed in Figure 6, the polarization resistance, *R*_pol_, i.e., the parameter characterizing the susceptibility of the material to corrosion can be calculated from the formula:(4)$$R_{\mathrm{pol}}=R_{p}+R_{b}$$

The Minuit program based on a complex nonlinear regression least-square fitting procedure was used to fitting the EEC model to the measured data [69]. The quality of the fit can be seen in Figure 7 and Figure 8, which show the example for the polished SMAT-2 sample using the EEC depicted in Figure 6.

Table 3 gathers the results of the most representative series of measurements. Three of such series were made, during which commonly observed variations in obtained results occurred. Differences in results of the corrosion measurements with nominally identical materials were reported by, e.g., op’t Hoog [70]. However, the obtained polarization resistances can be arranged as follows *R*_pol,reference_ < *R*_pol,SMAT-1_ < *R*_pol,SMAT-2_ for the as-received samples and *R’*_pol,reference_ < *R’*_pol,SMAT-1_ < *R’*_pol,SMAT-2_ for the polished samples. The SMAT process strongly influenced the *R*_pol_ of titanium. The porous layer resistance, *R*_p_, as well as, the inner barrier layer resistance, *R*_b_, increase with the increase in sonotrode amplitude and SMAT duration for the as-received SMATed samples. For the polished samples, *R*_b_ also increases while *R*_p_ does not change. However, for all samples, the values of porous layer resistance, *R*_p_, are negligible in comparison to the inner barrier layer resistance, *R*_b_, then they have a marginal effect on the polarization resistance, *R*_pol_.

Polishing of the samples led to an overall increase in corrosion resistance. This is understandable because polishing resulted in a smoother surface finish in comparison to the as-received samples which resulted in the reduction of the effective surface area that was exposed to the electrolyte during the EIS tests. Polishing allowed us to obtain surfaces in the same state, with roughly the same thickness of the oxide layer before the immersion in the electrolyte. This is reflected in almost the same values of the *R*_p_ calculated for the polished SMATed and the reference samples.

A significant influence of SMAT on the corrosion behavior of cp titanium in the 3.5 wt.% NaCl solution behavior was reported by Fu et al. [71]. Those authors attributed the enhanced corrosion resistance of SMATed titanium to the formation of a dense passive film on their surfaces. Additionally, a passive film that was formed on the SMATed specimens is more stable in comparison to the one formed on the non-treated sample, which makes it harder to cross for chloride ions. An increase in corrosion resistance USP cp titanium in the electrolyte similar to this applied in the present study, i.e., Ringer’s Solution was reported by Agrawal et al. [51]. Those authors attributed better corrosion resistance to the enhanced formation of a passive oxide protective film due to the presence of many grain boundaries on the surface of USP titanium. On the other hand, Jindal et al. [72] reported a drastic decrease of corrosion resistance of USP cp titanium in Ringer’s Solution in comparison to the non-treated specimen. A decrease in the corrosion resistance of peened titanium, which occurred despite nanostructured surface, was linked by these authors with inhomogeneity in dislocation density and residual stresses at the surface which hindered the formation of the uniform oxide layer. Generally, different corrosion responses can arise from different surface roughness, different densities of crystal lattice defects, and the magnitude of residual stresses. Hence, in the case of SMAT or USP, the peening parameters have a great influence on the corrosion resistance of the treated material.

Despite the rougher surface (specified by the parameter *R*_a_), the as-received SMAT-2 sample turned out to be more corrosion resistant than the SMAT-1 one. Increasing the surface area of the electrode connected to a higher roughness should be associated with a decrease in the measured resistance. It seems that both as-received and polished specimens owe their better corrosion resistance, in comparison to the corresponding references, to a larger number of nucleation sites for passive film formation, i.e., grain boundaries, dislocations, and twins. The presence of such structural inhomogeneities was confirmed by the EBSD picture and the PALS measurements. The better corrosion resistance of the as-received and the polished samples after SMAT-2 treatment, in comparison to the SMAT-1, coincides also with a slightly higher value of the shorter positron lifetime and its slightly higher intensity which is related to a higher dislocation density.

The obtained results indicate the positive effect of SMAT on the corrosion resistance of the titanium samples. Increasing simultaneously both the vibration amplitude and SMAT duration causes an increase of polarization resistance of titanium which resulted in its more noble corrosion behavior. The results show also that even after removal of the original oxide layer formed during contact with air, the microstructure that was created in the samples promotes the formation of a tight barrier layer on their surface. This, in turn, results in superior corrosion resistance in comparison to the corresponding reference sample.

## 4. Conclusions

The gradient effect of SMAT on the microstructure is reflected in the decrease in the microhardness values with the increasing distance from the treated surface. This effect is more apparent for the sample for which the energy imparted into the treated surface was higher. The application of the EBSD technique for this sample revealed the gradient microstructure. As the distance from the treated surface increases, the area rich in refined grains is followed by small grain clusters formed along the lines inclined relative to the surface.

The observed significant increase in the mean positron lifetime values points out many lattice defects introduced by SMAT. Two positron lifetime components were resolved in the positron lifetime spectra measured. The longer lifetime indicates the presence of vacancy clusters containing about 3 or 4 vacancies, while the shorter one originates from the annihilation of positrons trapped at dislocations.

The decrease of the shorter positron lifetime with the increasing distance from the treated surface indicates a decreasing dislocation density which was confirmed by a decrease of the microhardness. In the region for which the dislocation density is significantly lower than at the treated surface, a layer with a higher concentration of vacancy clusters indicated by the maximum of the longer lifetime intensity is observed. The increase of the energy imparted into the treated surface, connected to a higher sonotrode amplitude and longer duration of SMAT, shifts this vacancy cluster layer deeper into the material which is connected to the increase of the thickness of the whole affected layer.

EIS measurements revealed the positive effect of the SMAT on the corrosion resistance of the titanium samples studied. The better corrosion resistance for the sample treated longer and with the higher sonotrode amplitude coincides with slightly higher values of the shorter positron lifetime and its intensity which can be related to a higher dislocation density.

## Figures and Tables

**Figure 1 materials-14-06347-f001:**
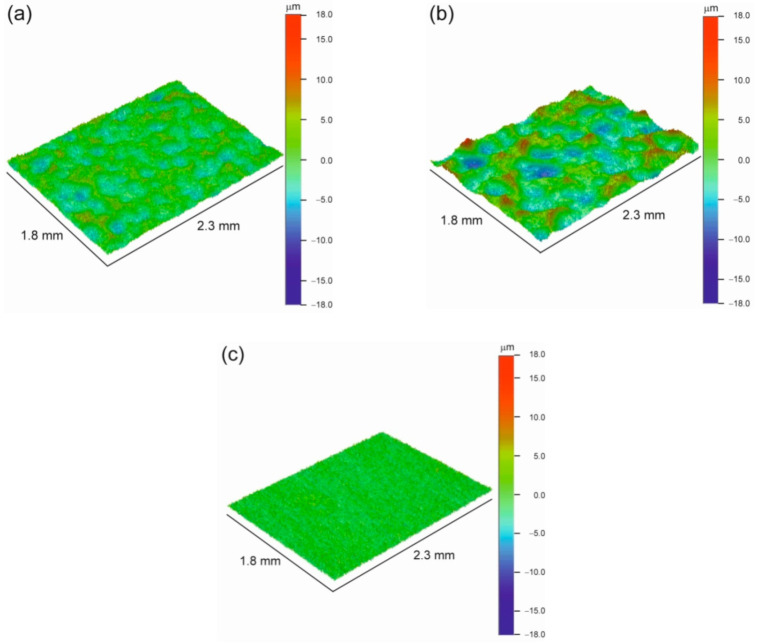
Optical profilometer images of the surface of the Ti specimens SMAT-1 (**a**), SMAT-2 (**b**), and the initial reference sample (**c**).

**Figure 2 materials-14-06347-f002:**
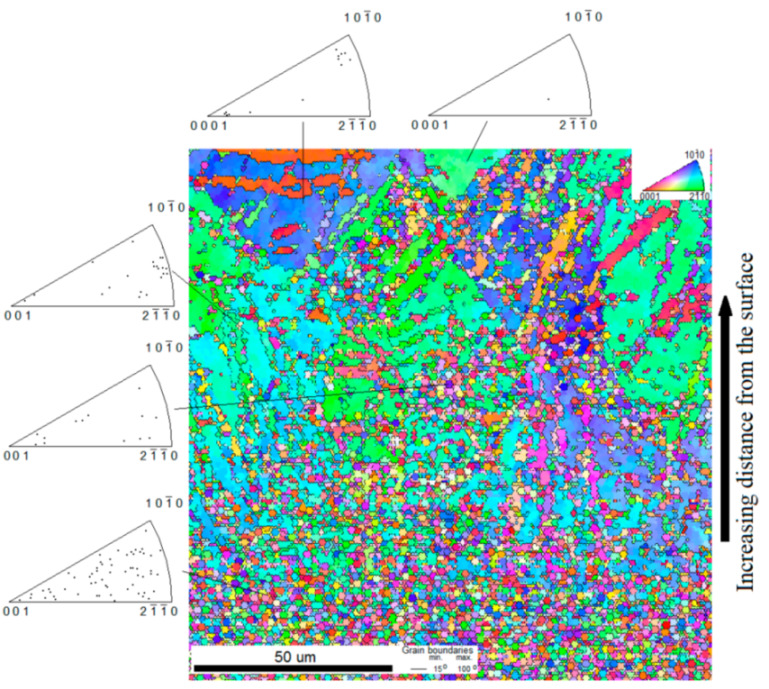
EBSD orientation mapping of the cross-section of SMAT-2 sample.

**Figure 3 materials-14-06347-f003:**
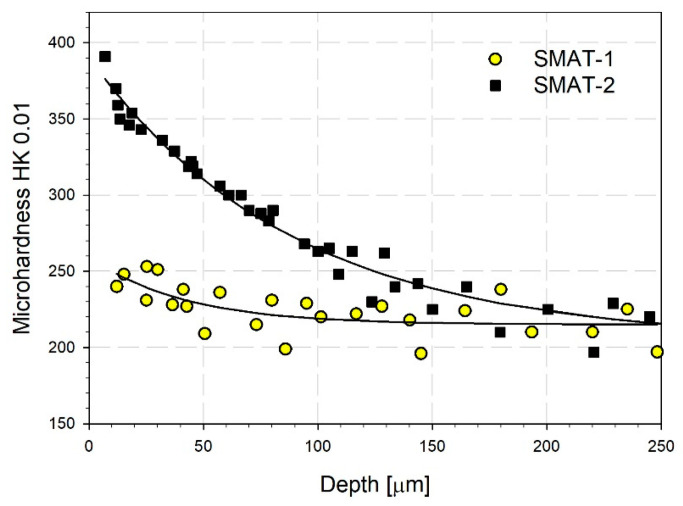
Variation of microhardness with distance from the treated surface. The solid lines represent the best fit of the exponential decay function to the experimental points: HK_SMAT-1_(z) = 215 + 44exp(−z/41) and HK_SMAT-2_ (z) = 204 + 186 exp(−z/89) for the samples SMAT-1 and SMAT-2, respectively; z is the distance from the treated surface.

**Figure 4 materials-14-06347-f004:**
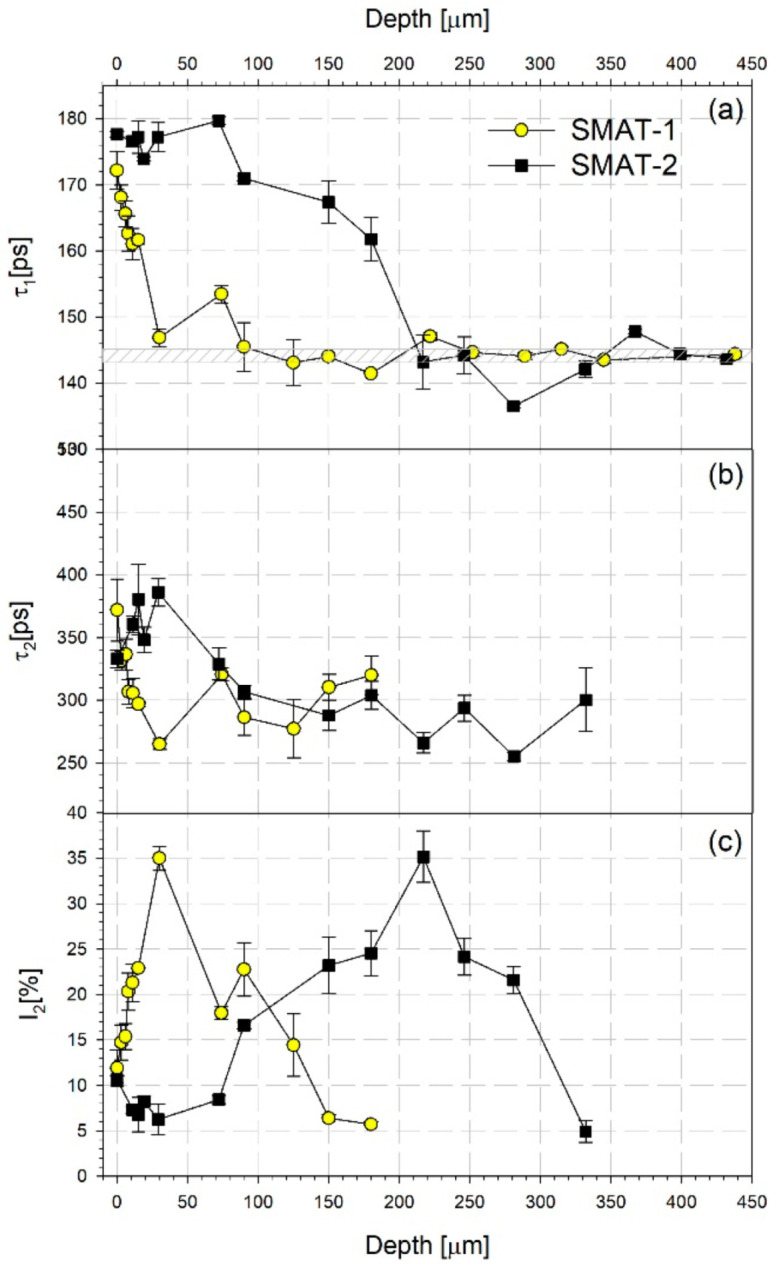
The depth profiles of the positron lifetime components, *τ*_1_ (**a**), *τ*_2_ (**b**), and intensity *I*_2_ (**c**), of the longer component obtained from the deconvolution of the positron lifetime spectra measured for the grade 2 titanium SMATed material.

**Figure 5 materials-14-06347-f005:**
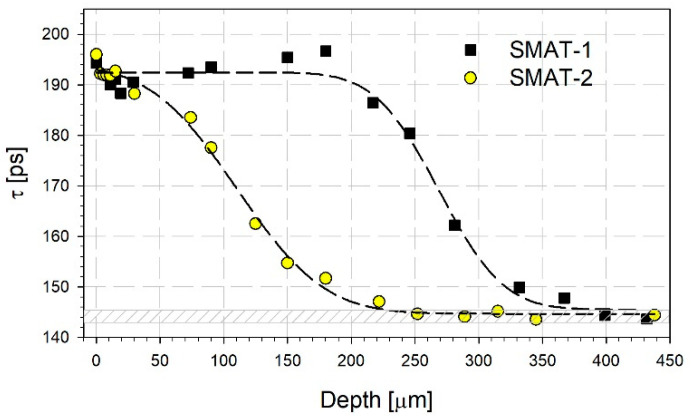
The depth profiles of the mean positron lifetime for the SMATed samples of grade 2 titanium. The dashed lines were obtained by fitting Equation (2) to the experimental points.

**Figure 6 materials-14-06347-f006:**
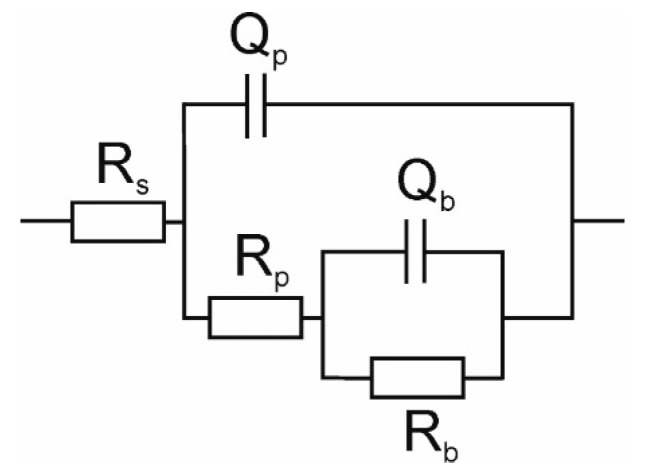
EEC applied for interpretation of the data from electrochemical impedance spectroscopy measurements.

**Figure 7 materials-14-06347-f007:**
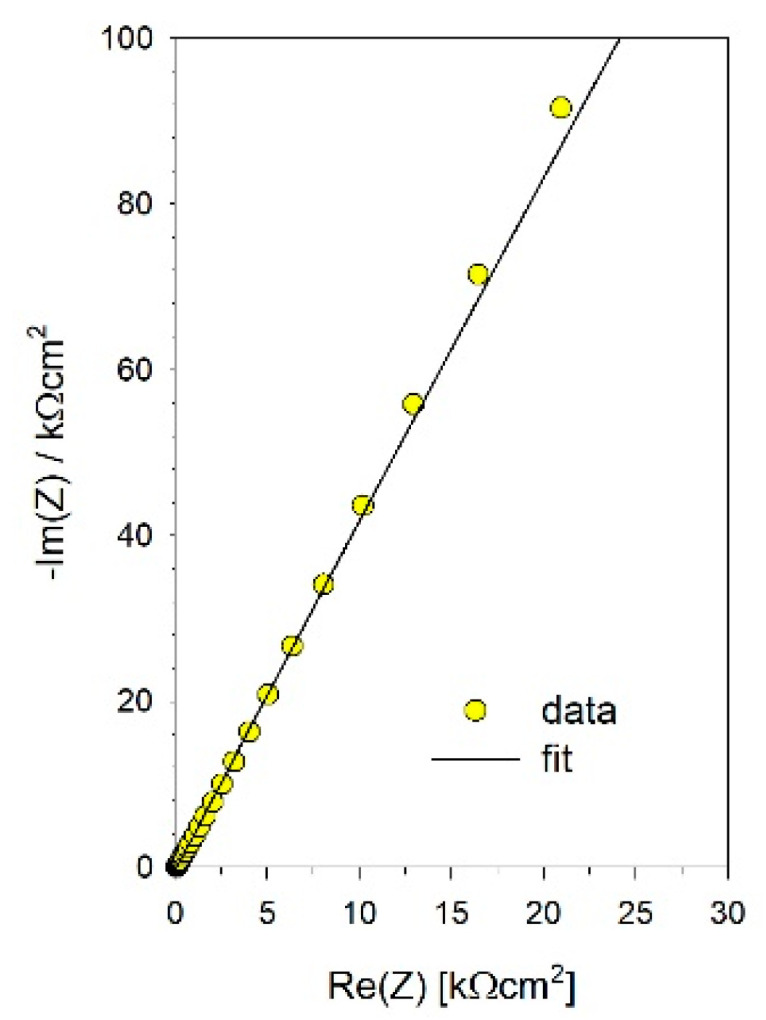
The example of the fit to the measured data for the polished SMAT-2 sample—Nyquist plot.

**Figure 8 materials-14-06347-f008:**
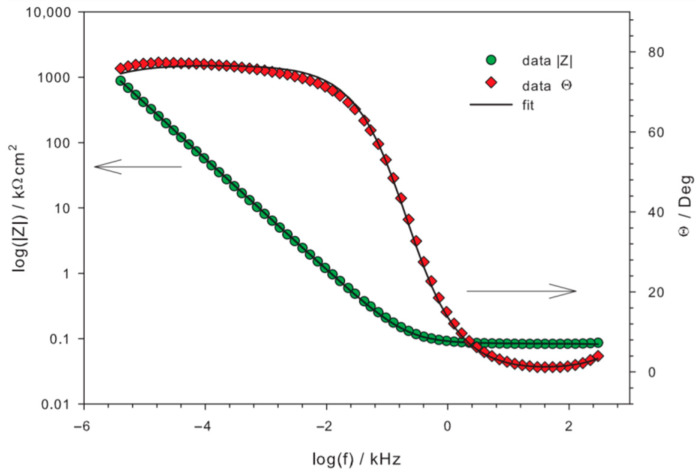
Example of the fit to measured data—Bode plots for the polished SMAT-2 sample.

**Table 1 materials-14-06347-t001:** The crystallite size and strain calculated using the Williamson-Hall method.

Sample	Crystallite Size [nm]	Microstrain [%]
SMAT-1	52 (5)	0.10 (1)
SMAT-2	68 (21)	0.28 (4)

**Table 2 materials-14-06347-t002:** The values of the adjusted parameters in Equation (2) used for the description of the obtained depth dependencies of the mean positron lifetime depicted in Figure 2.

Adjusted Parameters	SMAT-1	SMAT-2
*τ*_0_ [πσ]	144.7 (0.7)	145.5 (1.5)
*a* [ps]	49.3 (1.4)	46.9 (1.8)
*z*_0_ [µm]	110.5 (3.3)	269.1 (5.0)
*b* [µm]	53.9 (4.9)	39.8 (6.9)
Approximate total depth [µm]	250	400

**Table 3 materials-14-06347-t003:** Results of the electrochemical experiments carried out in 0.15 mol dm^−3^ NaCl solution on Ti samples.

Sample	*R*_p_ [Ωcm^2^]	*R*_b_ [MΩcm^2^]	*R*_pol_ [MΩcm^2^]
reference	143.7	1.53	1.53
SMAT-1	167.2	4.84	4.84
SMAT-2	190.1	5.29	5.29
reference polished	181.5	8.61	8.61
SMAT-1 polished	180.4	13.6	13.6
SMAT-2 polished	179.5	22.1	22.1

## Data Availability

The data presented in this study are available on request from the corresponding author.

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
