# Peer review of "Gradient Microstructure Induced by Surface Mechanical Attrition Treatment in Grade 2 Titanium Studied Using Positron Annihilation Spectroscopy and Complementary Methods"

_materials, 2021, doi:10.3390/ma14216347_

Round 1
Reviewer 1 Report
This is a very good informative paper focusing on microstructure in titanium using surface mechanical attrition treatment.
A good review here is available and I encourage authors to address some of the good point from the below references.
https://scholar.google.com/citations?view_op=view_citation&hl=en&user=LywUx3kAAAAJ&sortby=pubdate&citation_for_view=LywUx3kAAAAJ:olpn-zPbct0C
https://scholar.google.com/citations?view_op=view_citation&hl=en&user=LywUx3kAAAAJ&sortby=pubdate&citation_for_view=LywUx3kAAAAJ:_B80troHkn4C
Thank you
Author Response
Dear Sir or Madam,
please find attached the feedback responses for your review.
Best regards,
Konrad Skowron

Reviewer 2 Report
This manuscript from Skowron et al. shows their study of the microstructure of grade 2 titanium subjected to SMAT, and finds the relation to the corrosion properties. They found positron lifetime and lattice defects. The decreasing of dislocation density was confirmed by decreasing the microhardness and shows a decrease of shorter positron lifetime with increasing distance from the treated surface. The authors also show the positive effect of the SMAT on the corrosion resistance of the titanium samples in the study. However, I suggest the authors consider addressing some minor concerns, listed below:
1. Figure 1a and 1b, I suggest authors to match the scale bars of these two figures, so readers can clearly see the deeper craters in SMAT-2 and SMAT-1 is more homogenous.
2. Even though the authors briefly mentioned the SMAT application in the introduction part. I suggest authors go into more details about the application of the SMAT (either in the introduction part, or the conclusion part), in this way it can attract more readers.
Author Response

(The authors gave the same response as above.)

Reviewer 3 Report
Overall appreciation: It is an interesting and well done research in the field of alloys superficial cold hardening without affecting the bulk properties. The study is well documented with relevant references. The experimental section is well organized and the results are well discussed. The conclusions are supported by the experimental results, but must be improved by adding a paragraph to conclude the gradient effect of SMAT on the microstructure and micro-hardness. There are some corrections to be effectuated according to detailed comments below:
Line 108: You must specify the Titanium producer or supplier (e.g. Company Name and Country).
Figure 1 – The optical profile for the Ti reference is missing. It must be added to Figure 1 to illustrate the surface topography changes induced by SMAT treatment compared to the reference, and to sustain the Ra values discussed in the text.
The XRD patterns for Ti reference, SMAT1 and SMAT2 are missing. You must display those XRD patterns in a figure to illustrate the peaks broadening due SMAT treatment. They are also necessary to sustain the data presented in Table 1.
Figure 2: EBSD maps for the Ti reference and SMAT1 are missing. They must be presented beside the map for SMAT2. Y axis legend has typewriting errors ,,Increaing” must be corrected as ,,Increasing”.
Ref 20: The names of some authors are wrong typed please correct them as follows: ,, Jelliti S.” and ,, Demangel C.”
Ref 21 is identical with Ref 29, please delete one of them and correct the text.
Ref 36 and Ref 37 are identical, please delete one of them and correct the text.
Ref 46 is identical with Ref 55, please delete one of them and correct the text.
Ref 71 – title is missing.
Author Response

(The authors gave the same response as above.)
